# Comparative Transcriptome Analysis Reveals Common and Developmental Stage-Specific Genes That Respond to Low Nitrogen in Maize Leaves

**DOI:** 10.3390/plants11121550

**Published:** 2022-06-10

**Authors:** Song Guo, Adnan Arshad, Lan Yang, Yusheng Qin, Xiaohuan Mu, Guohua Mi

**Affiliations:** 1Institute of Agricultural Resources and Environment, Sichuan Academy of Agricultural Sciences, Chengdu 610066, China; guosong1999@163.com (S.G.); shengyuq@126.com (Y.Q.); 2College of Resources and Environmental Science, China Agricultural University, Beijing 100193, China; ad@cau.edu.cn; 3PODA Organization, Islamabad 44000, Pakistan; 4College of Resources and Environmental, Hunan Agricultural University, Changsha 410128, China; yanglan@hunau.edu.cn; 5Synergetic Innovation Center of Henan Grain Crops, College of Agronomy, Henan Agricultural University, Zhengzhou 450002, China; mulongyan2008@163.com; 6National Academy of Agriculture Green Development, Key Laboratory of Plant-Soil Interactions of MOE, China Agricultural University, Beijing 100193, China

**Keywords:** coexpression modules, division zone, elongation zone, maturation zone, RNA-seq

## Abstract

A growing leaf can be divided into three sections: division zone, elongation zone, and maturation zone. In previous studies, low nitrogen (LN) inhibited maize growth and development, especially leaf growth; however, the gene expression in response to LN in different regions in leaf were not clear. Here, using hydroponics and a transcriptome approach, we systematically analyzed the molecular responses of those zones and differentially expressed genes (DEG) in response to LN supply. Developmental stage-specific genes (SGs) were highly stage-specific and involved in distinct biological processes. SGs from division (SGs–DZ) and elongation zones (SGs–EZ) were more related to developmentally dependent processes, whereas SGs of the maturation zone (SGs–MZ) were more related to metabolic processes. The common genes (CGs) were overrepresented in carbon and N metabolism, suggesting that rebalancing carbon and N metabolism in maize leaves under LN condition was independent of developmental stage. Coexpression modules (CMs) were also constructed in our experiment and a total of eight CMs were detected. Most of SGs–DZ and SGs–EZ were classified into a set termed CM turquoise, which was mainly enriched in ribosome and DNA replication, whereas several genes from SGs–MZ and CGs were clustered into CM blue, which mainly focused on photosynthesis and carbon metabolism. Finally, a comprehensive coexpression network was extracted from CM blue, and several maize *CONSTANS-LIKE*(*ZmCOL*) genes seemed to participate in regulating photosynthesis in maize leaves under LN condition in a developmental stage-specific manner. With this study, we uncovered the LN-responsive CGs and SGs that are important for promoting plant growth and development under insufficient nitrogen supply.

## 1. Introduction

As one of the most important nutrient elements, nitrogen (N) plays a vital role in plant growth and development because it is a fundamental component of amino acids, proteins, nucleic acids, and many other important metabolites [1,2,3]. Ample amount of N fertilizer is used to maximize yields in crop production systems. Generally, only 40–50% of the applied N fertilizer is utilized by plants [4]. The N lost from the plant-soil system can result in environmental problems, including water and air pollution [5]. Thus, it is critical to increase crop production with less N fertilizer input.

Maize (*Zea mays* L.) is one of the most important food and feed crops worldwide [6]. Grain yield depends on the postsilking biomass accumulation, which is produced by photosynthesis and depends on the illuminated leaf area and the period of photosynthesis [7,8]. Previous studies have shown that low N (LN) stress reduces leaf area and photosynthesis and shortens the period of photosynthesis as LN promotes the remobilization of N from leaves, and thus reduces grain yield [2,9,10]. N deficiency stress reduces the expression levels of antenna system genes, light absorption, and transport in maize [9]. Reducing leaf area under LN could be due to both a lowering of leaf appearance rate and to a reduced leaf expansion by decreasing cell production rate and final cell length [9,11,12,13].

According to the state of cell movement, an expanding leaf could be divided into three sections: division zone (DZ), elongation zone (EZ), and maturation zone (MZ) [9,14]. Division zone encompasses several centimeters at the leaf base where cells proliferate [9,15]. In elongation zone, cells undergo elongation. Together, the two zones form the leaf growth zone. Cell proliferation and elongation have a close relationship with hormones such as auxin, cytokinins, and gibberellins [14,15,16,17]. When cells stop elongation, they enter into the maturation zone. Previous studies have reported that the differential gene-expression response to LN stress in the division zone is mainly enriched in genes involved in biological regulation, biosynthetic process, cellular biosynthetic process, response to abiotic stimulus, and cell homeostatic processes [9]. In the elongation zone, the differentially expressed genes focus on translation or organelle formation [9]. In maturation zone, the differentially expressed genes in response to low N stress are mainly involved in photosynthesis, homeostatic processes, regulation of nitrogen metabolism [2]. These are consistent with the function of cells in those positions.

Although the relationship between N deficiency and maize growth has been widely studied, the response of common genes (CGs) and developmental stage-specific genes (SGs) to LN in different positions of a developing maize leaf are not fully understood [2,9,18,19]. In the present study, we applied a comprehensive comparative transcriptome analysis of different leaf zones suffering from LN; the common and specific molecular functions underlying the strategies deployed within the maize leaves under LN were uncovered. The information obtained here will provide a molecular base for potential strategies of gene manipulation to promote plant growth and development under insufficient N supply.

## 2. Results and Discussion

### 2.1. Identification of LN-Responsive Genes in Different Leaf Regions

The PCA plots clearly separated the samples from different developmental stages and treatments; PC1 accounted for ca. 42% of total variation, whereas PC2 accounted for 18% (Figure 1A). The effects of developmental stages were uncovered by PC1, and samples from LN and high N (HN) were separated by PC3, which only occupied ca. 14% of total variations (Figure 1A). PCA plots clearly separated the replicates of different stages and treatments, suggesting good reliability of our RNA-seq data. Consistent with the PCA results, only ca. 3.2% (170 out of 5337) LN-responsive genes were commonly shared by the three different comparisons—i.e., DZ, EZ, and MZ—despite that there were 1582, 3112, and 1724 genes identified as LN-responsive genes in division zone, elongation zone, and maturation zone, respectively (Appendix A and Figure 1B). These 170 genes shared by three different comparisons were named common genes (CGs) for reference in further analysis (Appendix A). About 17–43% of the identified LN-responsive genes in maize leaves were stage-specific (Figure 1B) and defined as developmental stage-specific genes (SGs); thus, they generated three types of SGs, namely, SG in division zone (SGs–DZ), SG in elongation zone (SGs–DZ), and SG in maturation zone (SGs–DZ). These data strongly suggest that the LN-responsive genes in the different leaf zones were stage-specific.

### 2.2. SGs Are Highly Stage-Specific and Involved in Distinct Biological Processes

The KEGG analysis of the three different SGs showed that the SGs in different leaf zones go through different gene reprogramming events (Figure 1C). For instance, only the term of flavonoid biosynthesis was shared by both SGs–DZ and SGs–MZ (Figure 1C). The SGs in the division zone were also enriched in plant hormone signal transduction, MAPK signaling pathway, and biosynthesis of unsaturated fatty acids, whereas pathways of ribosome, DNA replication, and biosynthesis of amino acids were overrepresented in SG from elongation zone (Figure 1C). Plant hormones—for instance, auxins, cytokinins, and gibberellins—have a great effect on cell proliferation and elongation [14,15,16,17]. MAPK signaling pathway regulates cell division and differentiation, and thus, plays a vital role in plant growth and development [20,21]. Ribosome biogenesis and DNA replication are essential for protein biosynthesis and play key roles in cell division and elongation [22]. These results are consistent with the fact that these two leaf zones undergo cell proliferation and elongation, where the phytohormones and ribosome biogenesis play essential roles in regulating these processes [9,14,15,23]. Compared with SGs–DZ and SGs–EZ, the results obtained from SGs–MZ were much more informative (Figure 1C). The majority of LN-responsive SGs in the maturation zone were enriched in many metabolism processes, such as the metabolism related to pyruvate, glutathione, sugar, and amino acids (Figure 1C). Consistent with the KEGG analysis, the GO term analysis also indicated a contribution of LN-responsive SGs in different biological processes. Likewise, the SGs–DZ was enriched in several development processes, such as leaf development and stomatal complex development, as well as several pathways related to hormones (Appendix A). The SGs–EZ underwent reprogramming of the pathways related to ribosome biogenesis and DNA replication (Appendix A). However, the genes of SGs–MZ were more related to metabolic pathways, such as cellular carbohydrate and polysaccharide metabolic processes as well as cellular amino acid catabolic process (Appendix A), suggesting rebalancing of N and C metabolism in this leaf zone.

Collectively, the pathway analysis data implied that the LN-responsive SGs in the three different leaf zones affected distinct biological processes, which were closely linked to their differences in developmental stages. The LN-responsive SGs in the division and elongation zones were more related to development and ribosome biogenesis, respectively, whereas the SGs in the maturation zone were enriched in metabolism processes, such as carbohydrates.

### 2.3. CGs Are Involved in Key Pathways That Are Important for N Uptake and Metabolism

Despite the small proportion of shared significant LN-responsive genes among the three different leaf zones, the CGs still provided valuable information regarding the conserved adaptive strategies of maize leaves deployed to overcome the shortages in soil available N. The gene expression levels of these CGs were globally inhibited by LN treatments in the three different leaf zones (Figure 2A). Furthermore, the functions of these genes were mainly enriched in pathways related to photosynthesis and N metabolism, which were uncovered by the MapMan and KEGG analysis (Figure 2B,C). The GO term analysis also deeply supported this conclusion, since many terms related to photosynthesis were highly enriched in CGs (Appendix A). The inhibited capacities of photosynthesis by LN were observed in many plant species [2,24,25,26,27,28,29], suggesting the close links between carbon and N metabolism. The N metabolism provides substrates essential for chlorophyll biosynthesis, while LN reduces chlorophyll content in plants due to breakdown of chlorophyll to release N for reuse [2,9,26,30,31].

Additionally, many genes related to N metabolism, and protein biosynthesis and degradation, were identified in CGs (Table 1). For instance, the genes involved in N primary metabolism, such as *GLUTAMINE SYNTHETASE 2* (*ZmGS2*), *NITRITE REDUCTASE 1* (*ZmNIR1*), and *NITRATE REDUCTASE 1* (*ZmNIA1*), were downregulated in response to LN condition in three leaf zones (Table 1). Seven genes encoding ribosomal proteins, including *40S RIBOSOMAL PROTEIN S9* (*ZmRPS9C*) and RPL9, were also downregulated under LN in the three leaf zones (Table 1), suggesting that the translation processes might be slowed down, which could be attributed to the shortages of N-related substrates, such as amino acids. Consistent with this, three genes encoding proteins involved in aromatic and branched chain amino acids had reduced mRNA levels (Table 1). Low N decreased expression of genes for several N metabolism processes, including reduced mRNA levels of several genes that participate in N and peptides transport, such as maize *AMINO ACID PERMEASE 8* (*ZmAAP8*), *NRT1/PTR FAMILY 6.2* (*ZmNPF6.2*), and *ZmNPF5.16* (Table 1). These data suggest that declined N transport and metabolism processes are independent of developmental stage, and these processes may be responsible, in part, for the reduction in N content in grains and, finally, reduced yields.

Consistent with the reduced capacities of N metabolism, several important transcription factors (TFs) related to N metabolism were found in the CGs list. For instance, maize *NIN-like protein 6* (*ZmNLP6*) was significantly inhibited by LN in the three leaf zones, together with several known TFs involved in N metabolism, such as maize *LOB domain-containing protein 37a* (*ZmLBD37a*), *ZmLBD37b,* and *HRS1 homolog2* (*ZmHHO2*/*NIGT1.2*). In Arabidopsis, *NLP6* acts as a master in the nitrate-Ca_2_^+^-NLP-mediated nitrate signaling pathway, which plays essential roles in the so-called primary nitrate response [32]. The *AtNLP6*TF activates hundreds of genes involved in N uptake and metabolism in presence of nitrate, and *LBD37* in Arabidopsis is the target of *AtNLP6* in response to nitrate [32,33]. *AtLBD37* is a well-known TF that participates in N metabolism and leaf morphogenesis [34,35]. Furthermore, *HYPERSENSITIVITY TO LOW PI-ELICITED PRIMARY ROOT SHORTENING 1* (*HRS1*/*NIGT1*) in Arabidopsis participates in integrating nitrate and phosphate signals [36,37], and also can interact with *AtNLP6* to regulate N uptake [36]. Consistent with this, five *HRS1* homologs in maize were identified in CGs list (Table 2), indicating the possible roles of these genes in regulating maize N metabolism in leaves. The identified TFs in CGs were closely linked to N metabolism, implying their important roles in regulating their downstream genes in maize leaf to adapt the LN condition.

Collectively, our data clearly demonstrated that the CGs were mainly enriched in carbon and nitrogen metabolic pathways (Table 1 and Table 2, Figure 3), suggesting that rebalance of C and N metabolism under LN condition was independent of developmental stage in maize.

### 2.4. LN-Responsive Genes Are Highly Coexpressed; CGs and SGs Are Assigned into Different Modules

Recently, coexpression network analysis has been widely applied to dissect the transcriptional networks that control plant adaptation to various environmental changes [38,39,40]. A gene regulation network was built using the weighted correlation network analysis (WGCNA) method to gain more information of the transcriptional network that controls adaptation of LN condition in maize leaves (Figure 4). A total of 5337 genes were identified as significant in at least one comparison and were used for the analysis; ca. 4002 genes remained after filtering (Appendix A). The whole gene sets were separated into eight coexpressed modules (CMs), color-coded for reference, with the CM grey set containing the genes that could not be assigned to other CMs (Appendix A and Figure 4A). The CMs brown and green were more correlated compared with other CM comparisons (Figure 4B), suggesting the close gene expression profiles and similar biological functions between these CMs.

The CM turquoise consisted of 1347 genes with overall downregulated gene expression levels and represented the largest module among these modules (Figure 5). KEGG analysis of CM turquoise revealed that the mentioned genes were enriched in the ribosome-related pathway and DNA replication, which was similar to the results obtained by SGs–DZ (Figure 2C and Figure 5), suggesting the roles of this CM in the control of ribosome biogenesis in a stage-specific manner. Consistent with this result, a large proportion of SGs–EZ was assigned to this module (Appendix A). The GO term analysis of CM turquoise also revealed that several cellular amino acid processes were highly enriched in this CM together with ribosome biogenesis (Appendix A), signifying the close relationships between amino acid metabolism and translation regulation processes. The shortages of amino acid substrates may affect ribosome biogenesis and create a further slowdown of protein biosynthesis.

In addition to this, CMs green and brown showed distinct gene expression profiles compared with CMs turquoise and blue; the genes assigned to CMs green and brown were globally stimulated by LN (Figure 5). Moreover, relatively numerous genes from SGs–DZ and –EZ were assigned into CM brown, and genes from CM green occupied a large proportion of SGs–DZ and SGs–MZ (Appendix A). Although no significant KEGG pathways were detected in these two CMs (Figure 5), the GO term analysis of CM green showed that it was enriched in several terms related to cell wall, such as plant-type cell wall organization or biogenesis, and cell wall biogenesis (Appendix A). In contrast, the genes assigned into CM red were highly overrepresented in the GO terms related to stresses, such as response to wounding, toxin metabolic process, response to bacterium, and response to water (Appendix A). The activation of the genes assigned into these two CMs highlighted the enhanced capacities of plants to respond to diverse biotic and abiotic stresses under LN condition. Indeed, the trade-off between growth and defense response has been widely observed in many plant species. The inhibition of growth by LN may lead to resource fluxes that enter into defense-related pathways rather than growth by producing more secondary metabolites, which help plants to overcome adverse conditions [23,31,41]. However, among CGs, only a few genes were assigned to these two CMs, suggesting that these processes are highly stage-specific.

### 2.5. CM Blue Plays Essential Roles in Coordinating Multiple Metabolism Processes

CM blue had the most members of CGs (Figure 5 and Appendix A), which were enriched in pathways related to photosynthesis, amino acids, and carbon metabolism; this is consistent with the results obtained from CGs (Figure 2, Figure 3 and Figure 4). Interestingly, SGs–MZ were more assigned to CM blue rather than CM turquoise (Appendix A), suggesting that the pathway reprogramming events of MZ were more similar to CGs than to other comparisons, and the biological events of SGs–MZ were less developmental-stage-dependent.

To gain more information on CM blue, a comprehensive coexpression network was extracted from CM blue with weighted values above 0.5; this network consisted of 358 genes with 11,677 edges (Figure 6A), suggesting high correlations between genes within this network. This coexpression network was enriched in several terms related to photosynthesis (Appendix A and Figure 6B). To better understand the possible regulatory mechanisms underlying this network, TFs in the network were identified (Figure 6A). Surprisingly, a total of 13 out of 15 identified TFs were significantly downregulated in MZ, whereas none of them were significantly changed at their mRNA levels in DZ (Figure 6). This is consistent with the fact that most SGs–MZ were assigned into this CM (Appendix A). Interestingly, four out of the thirteen significant genes in MZ were identified as containing the domains CONSTANS (CO), CO-LIKE, or TOC1 (CCT)—namely, maize *CONSTANS-LIKE* 5 (*ZmCOL5*), *ZmCOL3*, *ZmCOL4,* and maize *B-BOX DOMAIN PROTEIN 15* (*ZmBBX15*) (Figure 6A). The CONSTANS proteins in Arabidopsis directly target the promoter of *FLOWERING LOCUS T* (*FT*), which regulates flowering time in angiosperms [42,43]. Thus, the downregulation of these four *ZmCOL* genes in leaf maturation zone might contribute to the LN triggered delay in flowering in maize, as observed in previous studies [44,45]. Furthermore, recent studies also uncovered the important roles of a chloroplast-localized CCT-domain protein in forming normal thylakoids and granum stacks in barley leaves, in which a defect in the barley *CCT MOTIF FAMILY 7* (*HvCMF7*) gene results in notable white stripes phenotypes [46,47]. The close correlation between CONSTANS-LIKE genes and chlorophyll accumulation have also been discovered in other plant species [48,49]. Thus, these *CCT*-domain-containing genes, which were identified in CM blue, might play possible roles in regulating photosynthesis and flowering time in maize leaves under LN condition in a stage-specific manner.

## 3. Materials and Methods

### 3.1. Plant Material and Growth Condition

Seeds of maize inbred line B73 were sterilized in 10% (*v*/*v*) H_2_O_2_ for 30 min, rinsed with distilled water and soaked in saturated CaSO_4_ solution for 8 h, and then transferred on to filter paper to germinate in the dark at room temperature. When the roots were approximately 1 cm-long, the seeds were transferred to rolled papers and cultured. The seedlings with two visible leaves were then transferred into porcelain pots (4 seedlings per pot) containing 2 L of nutrient solution after the endosperm was removed. The plants were cultured in a growth chamber at 28/22 °C (day/night) with light intensity of 250–300 µmol of photons (400–700 nm) m^−2^ s^−1^ and a 14/10-h light/dark cycle. The pots were arbitrarily placed and rotated when the nutrient solution was renewed.

The nutrient solution contained 0.75 mM K_2_SO_4_, 0.1 mM KCl, 0.25 mM KH_2_PO_4_, 0.65 mM MgSO_4_, 0.13 mM EDTA-Fe, 1.0 µM MnSO_4_, 1.0 µM ZnSO_4_, 0.1 µM CuSO_4_, and 0.005 µM (NH_4_)_6_Mo_7_O_24_ [50]. Plants were supplied with half strength nutrient solution for 2 d and then transferred into full-strength solution with 4.0 mM NO_3_^−^ (high nitrogen; HN). When the third leaf was fully expanded (approximately 6 days later, V3 stage), half of the plants were moved into a solution with 40 μM NO_3_^−^ (low nitrogen; LN), with CaCl_2_ added to equalize calcium concentration between the treatments. The pH of the solution was adjusted to 5.8–6.0 with KOH. The nutrient solution was renewed every other day and aerated continuously by a pump.

### 3.2. RNA Library Construction and Illumina Sequencing

At leaf V3 stage, 160 plants and 240 plants were treated with high nitrogen and low nitrogen, respectively. Leaf segments of 0–5 mm (distance from leaf base), 25–45 mm (distance from leaf base), and 50–150 mm (distance from tip) of the sixth leaf were sampled, representing division region, elongation region, and maturation region, respectively [51]. The samples harvested from different N treatments and leaf zones were named as LN division zone (LDZ), HN division zone (HDZ), LN elongation zone (LEZ), HN elongation zone (HEZ), LN maturation zone (LMZ), and HN maturation zone (HMZ). There were two biological replicates for each treatment. The samples were frozen in liquid N_2_ and stored at −80 °C. Total RNA was extracted as described by Gu et al. [52]. RNA fragments were reverse-transcribed to create the final cDNA library in accordance with the protocol for the mRNA-Seq sample preparation kit (Illumina). The average insert size for the paired-end libraries was 125 bp. Paired-end sequencing was performed on an IlluminaHiSeq platform (IlluminaHisSeq 2500).

### 3.3. Alignment and the Identification of Differentially Expressed Genes (DEG)

Raw reads were preprocessed to remove low-quality reads and adaptor sequences. The clean reads of each sample were used to align the reads to the maize reference genome (B73 RefGen_V4, https://phytozome.jgi.doe.gov/pz/portal.html, accessed on 6 January 2020) using subjunc function in the Subread software [53]. Then, the obtained BAM files were subjected to feature Counts software [54], to obtain count information of the detected genes in each sample. Furthermore, the differentially expressed genes were calculated with the “DESeq2” packages [55] in R (https://www.r-project.org/, accessed on 6 January 2020) using count information of each gene; the genes with fold changes above 2 and false discovery rate (FDR) lower than 0.05 were selected as significantly differentially expressed genes. The fragments per kilobase of transcript per million reads (FPKM) of each gene were calculated for further use.

### 3.4. Bioinformation Analysis

The maize genes were annotated with the annotation file from Phytozome (http://www.phytozome.net/, accessed on 13 January 2020), and the closest Arabidopsis homologs of each maize gene was obtained; the symbols and descriptions of Arabidopsis genes were obtained from TAIR (Araport11, https://www.arabidopsis.org/, accessed on 14 January 2020).

To determine the reliability of RNA-seq data, principal component analysis (PCA) was conducted with “FactoMineR” package [56] in R. The scores of each sample in the first three principal components (PCs) were obtained and the 3D scatterplot of PCA was drawn in OriginPro software (version 2015, Origin Lab Corporation, Northampton, USA).

The Venny software (https://bioinfogp.cnb.csic.es/tools/venny/, accessed on 19 January 2020) [57] was used to find the common and specific LN-responsive gene sets among the three comparisons, i.e., LDZ vs. HDZ (DZ), LEZ vs. HEZ (EZ), and LMZ vs. HMZ (MZ). The shared genes by the three comparisons were defined as common genes (CGs), whereas the unique significant genes for just one comparison were defined as development stage-specific genes (SGs), thus generating three types of SGs, namely, SGs–DZ, SGs–EZ, and SGs–MZ. The transcription factors in CGs were identified based on the list obtained from PlantTFDB (http://planttfdb.cbi.pku.edu.cn/index.php, accessed on 12 January 2020) [58].

The MapMan analysis of CGs was conducted in BAR database with the Classification SuperViewer Tool [59] using the corresponding Arabidopsis homologous genes. The gene ontology (GO) and Kyoto Encyclopedia of Genes and Genomes (KEGG) analysis were accomplished in R with the packages of “clusterProfiler” [60]. For GO term analysis, the significant GO term assigned into biological process (BP) was further subjected to REViGO (http://revigo.irb.hr/, accessed on 17 January 2020) to redundant GO terms, the results were further visualized in Cytoscape software, University of California-San Diego, La Jolla, USA (version 3.7.1) [61]. Bubble plots were drawn using “ggplot2” packages [62] in R to display the results of GO and KEGG analysis, as suggested by Luo, Xia, Cao, Xiao, Zhang, Liu, Zhan, and Wang [38]. The heatmap was generated in R with the “pheatmap” packages [63] following the methods of Luo, Liang, Wu, and Mei [40].

### 3.5. Coexpression Network Analysis

The pooled gene list of DEG from the three comparisons (i.e., DZ, EZ, and MZ) was used for coexpression analysis. The FPKM value of each gene in different samples was used for coexpression analysis using the weighted correlation network analysis (WGCNA) method in R with the “WGCNA” packages [64]. The soft threshold was determined whereby more than 85% of the models would fit to scale-free topology and low mean connectivity. The minimum gene number assigned into one module was set as 30. The network of coexpressed module (CM) was visualized in Cytoscape software, University of California-San Diego, La Jolla, USA (version 3.7.1) [61].

## 4. Conclusions

In conclusion, during the early vegetative stage, large sets of LN-responsive genes were identified in different maize leaf zones. The comparative transcriptome analysis revealed that most of the LN-responsive genes were stage-specific; the LN-responsive genes from the three different leaf zones only shared a very small proportion. The stage-specific genes of the different zones were enriched in different pathways: SGs–DZ and SGs–EZ were more related to developmental dependence, whereas SGs–MZ were more related to metabolic processes, similar to the enriched pathways of CGs. The CGs were overrepresented in carbon and N metabolism, suggesting that rebalancing carbon and N metabolism in maize leaves under LN condition was independent of developmental stage. Furthermore, the coexpression network analysis assigned most SGs and CGs into different CMs; large numbers of SGs–DZ and SGs–EZ were assigned into CM turquoise, whereas several genes from SGs–MZ and CGs were clustered into CM blue. The functions of two activated CMs clearly related to cell wall and defense responses showed developmental stage-specificity. Finally, a comprehensive coexpression network was extracted from CM blue, and several *ZmCOLs* seemed to participate in regulating the photosynthesis in maize leaves under LN condition in a stage-specific manner.

## Figures and Tables

**Figure 1 plants-11-01550-f001:**
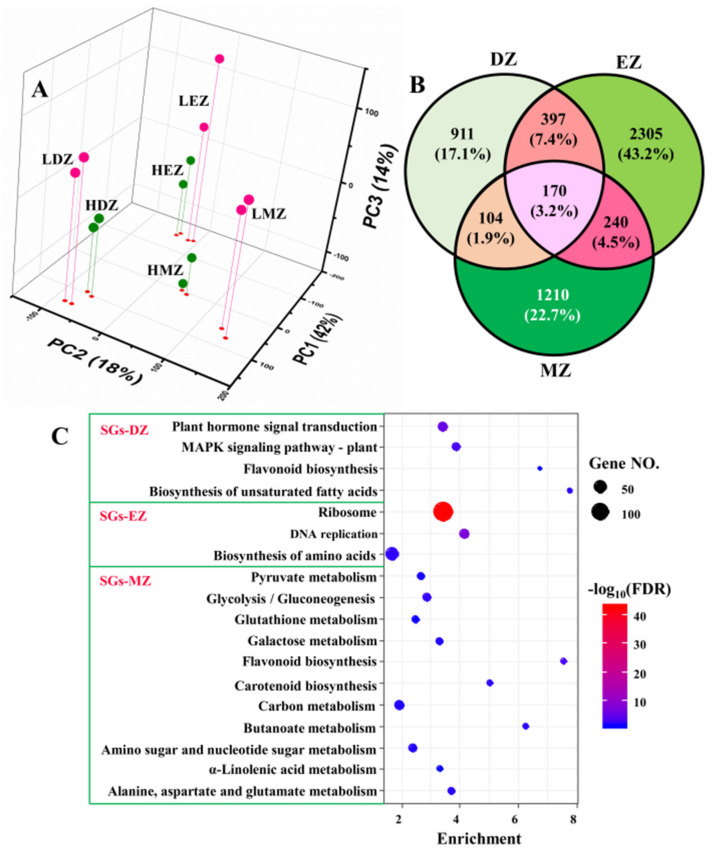
The 3D scatterplot represents the first three principal components (PCs) of the gene expression profiles in different regions of maize leaves under low nitrogen (LN) and high nitrogen (HN) conditions (**A**). LDZ—division zone under LN; HDZ—division zone under HN; LEZ—elongation zone under LN; HEZ—elongation zone under HN; LMZ—mature zone under LN; HMZ—mature zone under HN. The Venn diagram shows the stage-specific genes (SGs) and common genes in the three different comparisons (**B**), namely, LDZ vs. HDZ (DZ for short), LEZ vs. HEZ (EZ), and LMZ vs. HMZ (MZ). In panel (**C**), the KEGG analysis of the stage-specific genes (SGs) in division zone (SGs–DZ), elongation zone (SGs–EZ), and mature zone (SGs–MZ) are shown.

**Figure 2 plants-11-01550-f002:**
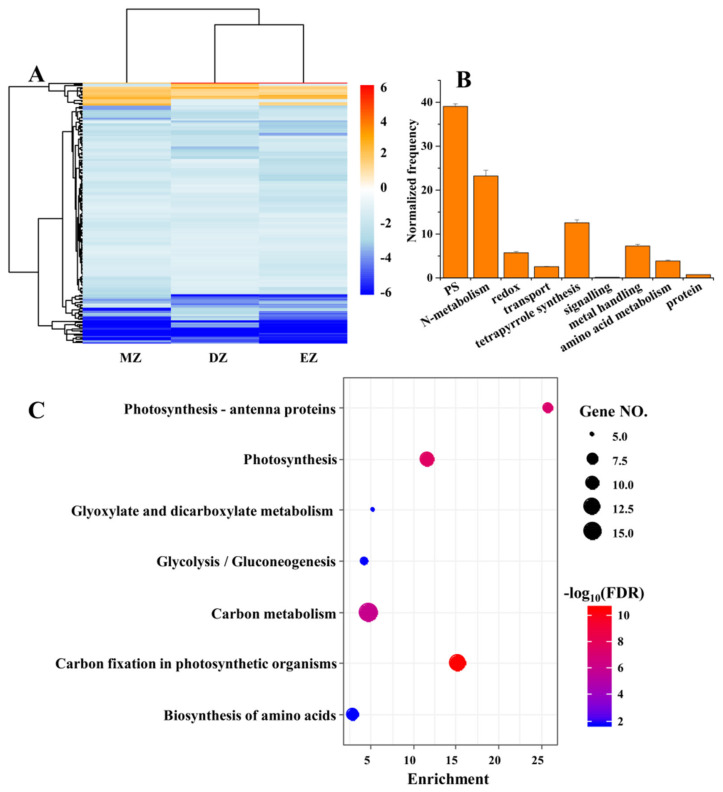
The heatmap represents the gene expression levels of common genes (CGs) under the comparisons of DZ, EZ, and MZ (**A**). The MapMan analysis (**B**) and KEGG (**C**) of CGs that are shared by the three comparisons. The MapMan analysis was conducted in BAR database (http://bar.utoronto.ca/welcome.htm, accessed on 10 January 2020) with the Classification SuperViewer tool; only the significant terms are shown in the figure.

**Figure 3 plants-11-01550-f003:**
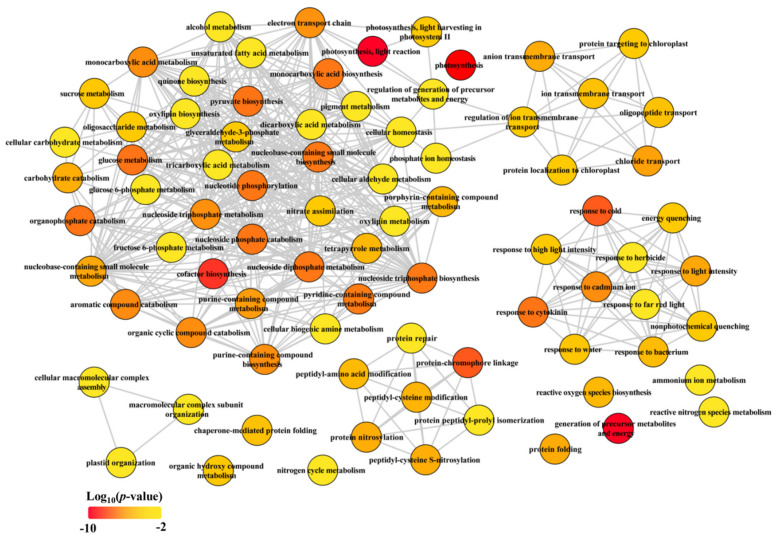
Biological process of GO term analysis of common genes (CGs). The GO term analysis was conducted in R with package “clusterProfiler”. The full list of significant GO terms (see Appendix A) assigned into biological process were subjected in REViGO (http://revigo.irb.hr/, accessed on 17 January 2020) to redundant GO terms and visualized in Cytoscape.

**Figure 4 plants-11-01550-f004:**
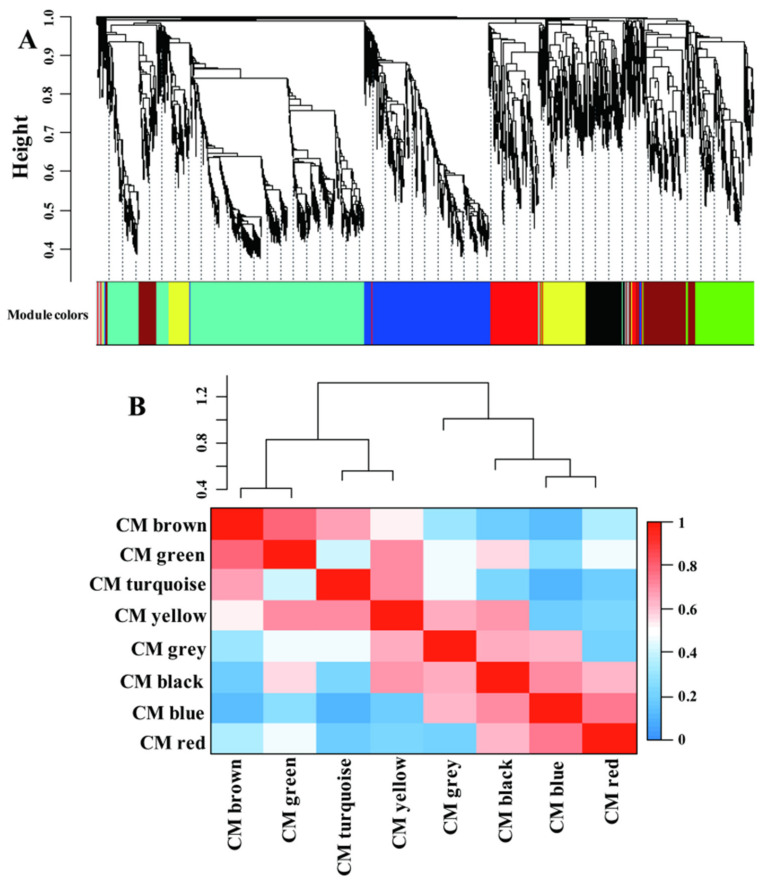
The significantly differential expressed genes assigned into different coexpression modules (CMs) by WGCNA method (**A**). The correlative relationships of the assigned CMs (**B**).

**Figure 5 plants-11-01550-f005:**
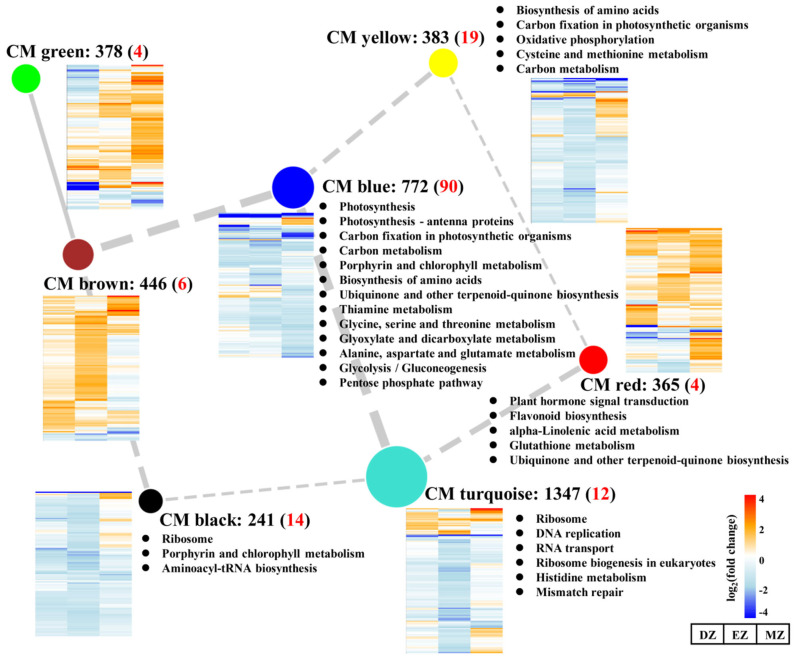
Network of correlated coexpression modules (CMs). The nodes indicate the different assigned CMs, while the edges represent the correlations between different CMs. The width of edges represents the correlation values between two CMs, and the solid and dashed edges represent positive and negative correlations, respectively. Only the absolute values of correlation values above 0.5 are shown in the figure. The black numbers behind the CM represent the number of genes assigned into the CM, and the red numbers in the brackets represent the number of common genes assigned into the CM. The heatmaps close to the CM represent the gene expression profiles assigned in the CM. The KEGG analysis of each CM is indicated.

**Figure 6 plants-11-01550-f006:**
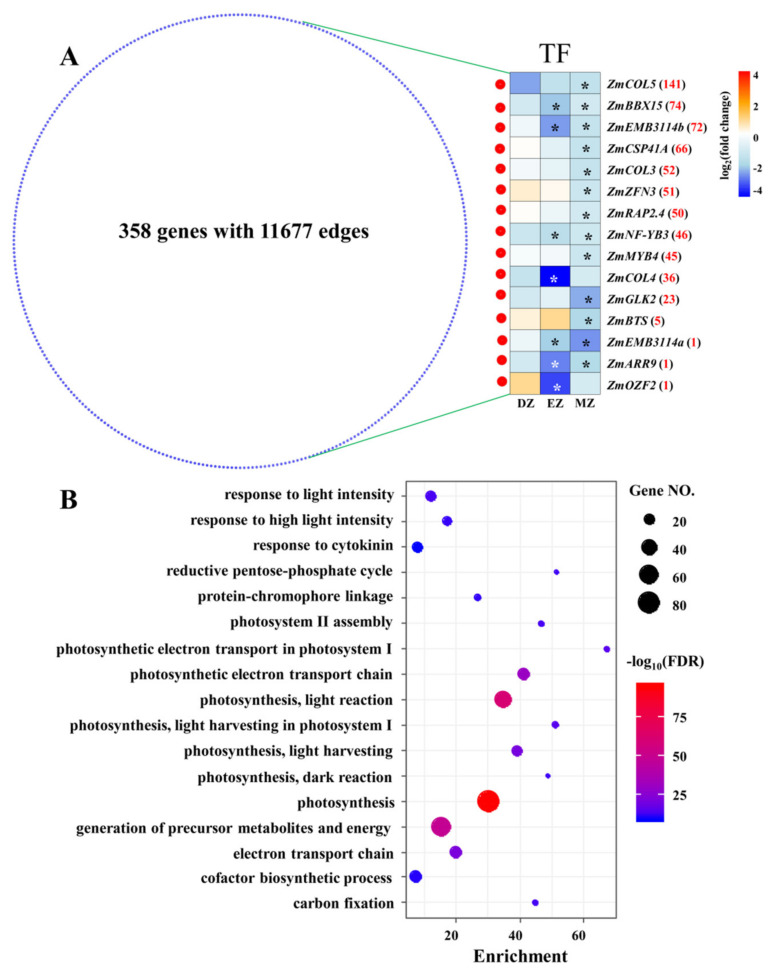
Network derived from CM blue with weighted value above 0.5 (**A**). The red and blue nodes represent transcription factors (TF) and other genes, respectively. The expression profiles of TF in the three comparisons are shown as a heatmap. The stars in the heatmap indicate the significance of genes under a certain comparison (FDR ≤ 0.05). The red numbers behind the gene name represent the number of coexpressed genes for each TF. COL, CONSTANS-LIKE; BBX15, B-BOX DOMAIN PROTEIN 15; EMB3114, EMBRYO DEFECTIVE 3114; CSP41A, CHLOROPLAST STEM-LOOP BINDING PROTEIN OF 41 KDA; ZFN3, ZINC FINGER NUCLEASE 3; RELATED TO AP2 4 RAP2.4, RELATED TO AP2 4; NF-YB3, NUCLEAR FACTOR Y, SUBUNIT B3; MYB4, MYB DOMAIN PROTEIN 4; GLK2, GOLDEN2-LIKE 2; BTS, BRUTUS; ARR9, RESPONSE REGULATOR 9; OZF2, OXIDATION-RELATED ZINC FINGER 2. The most significant biological process of GO terms for the genes that appeared in the network (**B**); the full list is given in Appendix A.

**Table 1 plants-11-01550-t001:** Genes involved in N-relative pathways in the common genes.

Locus Name	Name *	Closet AGI	Symbol of AGI	Description	DZ	EZ	MZ
**Amino acid and N metabolism**							
Zm00001d050694	*ZmEMB1075*	AT1G43710	*EMB1075*	embryo defective 1075	−2.1	−3.0	−1.9
Zm00001d052247	*ZmSK1a*	AT2G21940	*SK1*	shikimate kinase 1	−1.6	−1.1	−1.2
Zm00001d018061	*ZmSK1b*	AT2G21940	*SK1*	shikimate kinase 1	−1.4	−1.6	−1.6
Zm00001d002880	*ZmIDM2*	AT1G80560	*IMD2*	isopropylmalate dehydrogenase 2	−1.1	−1.8	−1.3
Zm00001d026501	*ZmGS2*	AT5G35630	*GS2*	glutamine synthetase 2	−1.1	−1.8	−1.4
Zm00001d018161	*ZmNIR1*	AT2G15620	*NIR1*	nitrite reductase 1	−5.8	−5.6	−2.9
Zm00001d049995	*ZmNIA1*	AT1G77760	*NIA1*	nitrate reductase 1	−2.8	−2.5	−2.5
**Protein synthesis and degradation**							
Zm00001d047186	*Zm* *RPS9C*	AT5G39850	*RPS9C*	40S ribosomal protein S9	−1.3	−2.6	−1.2
Zm00001d038084	*ZmEMB3126*	AT3G63490	*EMB3126*	embryo defective 3126	−1.1	−1.4	−1.0
Zm00001d016072	*ZmRPL21C*	AT1G35680	RPL21C	chloroplast ribosomal protein l21	−1.9	−1.5	−1.4
Zm00001d014488	*ZmRPL24*	AT5G54600	*RPL24*	plastid ribosomal protein l24	−1.4	−1.2	−1.4
Zm00001d044130	*ZmEMB2184*	AT1G75350	*EMB2184*	embryo defective 2184	−1.0	−1.6	−1.3
Zm00001d018412	*ZmRPL9*	AT3G44890	*RPL9*	ribosomal protein l9	−1.7	−1.3	−1.6
Zm00001d015628	*ZmGHS1*	AT3G27160	*GHS1*	glucose hypersensitive 1	−1.8	−2.2	−1.9
Zm00001d043195	*ZmMC5*	AT1G79330	*MC5*	metacaspase 5	−1.1	−1.0	−2.0
Zm00001d046952	*ZmAAH*	AT4G20070	*AAH*	allantoateamidohydrolase	1.3	1.4	1.8
**Transport related to N**							
Zm00001d012231	*ZmAAP8*	AT1G10010	*AAP8*	amino acid permease 8	−4.3	−4.3	−3.3
Zm00001d044529	*ZmNPF6.2*	AT2G26690	*NPF6.2*	NRT1/ PTR family 6.2	−6.8	−8.4	−8.3
Zm00001d042684	*ZmNPF5.16*	AT1G22550	*NPF5.16*	NRT1/ PTR family 5.16	−6.1	−8.4	−8.3
Zm00001d051525	*ZmOPT4*	AT5G64410	*OPT4*	oligopeptide transporter 4	1.8	2.3	2.7
Zm00001d045519	*ZmOPT6*	AT4G10770	*OPT7*	oligopeptide transporter 7	2.3	2.3	2.2
Zm00001d015702	*ZmCLCG*	AT5G33280	*CLCG*	chloride channel G	−2.1	−1.1	−1.7
Zm00001d046919	*ZmCLC-Aa*	AT5G40890	*CLC-A*	chloride channel A	−1.8	−2.9	−1.5
Zm00001d015700	*ZmCLC-Ab*	AT5G49890	*CLC-A*	chloride channel A	−2.6	−1.8	−1.6

* The names of maize genes were given based on their closet Arabidopsis homologue. AGI—Arabidopsis Gene ID; DZ—division zone; EZ—elongation zone; MZ—maturation zone.

**Table 2 plants-11-01550-t002:** Transcription factors identified in the common genes.

Locus Name	Name *	Closet AGI	Symbol of AGI	Description	DZ	EZ	MZ
Zm00001d034160	*ZmHHO5*	AT4G37180	*HHO5*	HRS1 homolog5	−4.8	−5.2	−2.7
Zm00001d013202	*ZmHHO4a*	AT2G03500	*HHO4*	HRS1 homolog4	−5.6	−5.5	−4.8
Zm00001d030891	*ZmHHO4b*	AT2G03500	*HHO4*	HRS1 homolog4	−2.0	−1.8	−3.1
Zm00001d007962	*ZmHHO4c*	AT2G03500	*HHO4*	HRS1 homolog4	−3.2	−1.5	−1.8
Zm00001d023402	*ZmHHO2*	AT1G68670	*HHO2; NIGT1.2*	HRS1 homolog2	−3.5	−7.3	−8.1
Zm00001d051749	*ZmHB-1*	AT3G01470	*HB-1*	homeobox 1	1.3	1.0	−1.2
Zm00001d029601	*ZmLBD37a*	AT5G67420	*LBD37*	LOB domain-containing protein 37	−3.8	−7.4	−5.5
Zm00001d021995	*ZmLBD37b*	AT5G67420	*LBD37*	LOB domain-containing protein 37	−6.0	−5.7	−8.1
Zm00001d018255	*ZmNF-YA5*	AT1G54160	*NF-YA5*	nuclear factor Y, subunit A5	1.5	2.7	2.0
Zm00001d015201	*ZmNLP6*	AT1G64530	*NLP6*	NIN-like protein 6	−3.9	−4.9	−4.1

* The names of maize genes were given based on their closet Arabidopsis homologue. AGI—Arabidopsis Gene ID; DZ—division zone; EZ—elongation zone; MZ—maturation zone.

## Data Availability

Raw sequencing data are stored at the Sequence Read Archive (Available online: http://www.ncbi.nlm.nih.gov/sra, accessed on 12 May 2022) under accession number GSE107562 and GSE111425.

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
