# Peer review of "Comparative Transcriptome Analysis Reveals Common and Developmental Stage-Specific Genes That Respond to Low Nitrogen in Maize Leaves"

_plants, 2022, doi:10.3390/plants11121550_

Round 1
Reviewer 1 Report
Dear Authors,
the manuscript I reviewed entitled: "Comparative transcriptome analysis reveals common and specific low nitrogen responsive genes indifferent sections of maize leaves" by Song Guo, Adnan Arshad, Lan Yang, Yusheng Qin, Xiaohuan Mu, Guohua Mi * presents the results of studies in which the Authors applied a comprehensive comparative transcriptome analysis of different maize leaf zones suffered from low nitrogen stress.
Ambitious and interesting scope of the research concerns one of the most important cultivated species - maize. The authors were able to identify common and specific molecular mechanisms that are important for promoting the growth and development in a nitrogen deficient setting.
The research presented in this manuscript has significantly expanded the database on gene manipulation regarding promoting plant growth and development in nitrogen deficiency conditions.
I have only a few minor reservations about this manuscript:
1) Keywords: the word "nitrogen" was used in the title of the paper, so it should not be reused as a keyword.
2) Abstract: there is no clear hypothesis here. In addition, the Abstract should mention the methods used in the research.
3) Introduction: the relationship between nitrogen deficiency and maize growth needs to be extended. In addition, please expand also the topic of similar molecular studies carried out for other species.
4) Tables 1 and 2: in the title or under the table, the abbreviations should be explained: AGI, DZ, EZ, MZ, because in general Tables and Figures should be understandable without the need to look for an explanation of the abbreviations in the text.
5) Figure 4: not marked Figures A and B.
6) The manuscript is quite neatly aligned with the Plants template, but still needs some tweaking.
Overall, I believe that the results of the research presented in this manuscript make a significant contribution to understanding the mechanisms of genetic regulation of maize crop stresses, and the Plants journal should consider publishing it.
Author Response
1) Keywords: the word "nitrogen" was used in the title of the paper, so it should not be reused as a keyword.
R1: Yes, We delete "nitrogen" as a keyword in revision.
2) Abstract: there is no clear hypothesis here. In addition, the Abstract should mention the methods used in the research.
R2: Thanks for your suggestion. We add "using hydroponics and transcriptome approach" in the abstract.
3) Introduction: the relationship between nitrogen deficiency and maize growth needs to be extended. In addition, please expand also the topic of similar molecular studies carried out for other species.
R3: Thanks for your suggestion. We add the following to the introduction : N deficiency stress reduced the expression levels of antenna system genes, light absorption and transport in maize [9]. Low-N stress may lead to tiller bud dormancy by altering the expression of cell cycle-related genes in rice [11].
4) Tables 1 and 2: in the title or under the table, the abbreviations should be explained: AGI, DZ, EZ, MZ, because in general Tables and Figures should be understandable without the need to look for an explanation of the abbreviations in the text.
R4: Yes, We explained AGI, DZ, EZ, MZ in notes.
5) Figure 4: not marked Figures A and B.
R5: Yes, We marked A and B in Figure 4 in revision.
6) The manuscript is quite neatly aligned with the Plants template, but still needs some tweaking.
R6: Yes, We adjusted the format according to the requirements of journals in revision.
Reviewer 2 Report
This article is a very innovative approach, with new insights that will certainly increase the understanding of N metabolism in monocotyledonous plants!

Author Response
We revised according to your comments marked on the manuscript. Details as follows:
1) Check English language and style.
2) Expand the studies of genes response to LN stress in introduction.
3) Modify Fig. to Figure.
4) Revise the conclusion.
Reviewer 3 Report
Manuscript "Comparative Transcriptome Analysis Reveals Common and Specific Low Nitrogen Responsive Genes Indifferent Sections of Maize Leaves" is interesting.
Authors applied a comprehensive comparative transcriptome analysis of different leaf zones suffered from LN, the common and specific molecular mechanisms underlying the strategies deployed within the maize leaves under LN were uncovered. Obtained information will provide a molecular base for gene manipulation regarding promoting plant growth and development under insufficient N supply.
Quality of Figure 1 is very poor.
Description of statistical analysis is not enough. Lack is information about distribution. After PCA Authors should use discriminant analysis.
Paper needs minor correction.
Author Response
1) Quality of Figure 1 is very poor.
R1:Yes,We submitted the HD original image of picture 1.
2) Description of statistical analysis is not enough. Lack is information about distribution. After PCA Authors should use discriminant analysis.
R2: Thanks for your suggestion. We added the following sentence in Results and Discussion: PCA plots clearly separated the replicated of different tissues and treatments, suggesting good reliability of our RNA-seq data.